# Learning Diverse Options via InfoMax Termination Critic

## Abstract

We consider the problem of autonomously learning reusable temporally extended actions, or options, in reinforcement learning. While options can speed up transfer learning by serving as reusable building blocks, learning reusable options for unknown task distribution remains challenging. Motivated by the recent success of mutual information (MI) based skill learning, we hypothesize that more diverse options are more reusable. To this end, we propose a method for learning termination conditions of options by maximizing MI between options and corresponding state transitions. We derive a scalable approximation of this MI maximization via gradient ascent, yielding the InfoMax Termination Critic (IMTC) algorithm. Our experiments demonstrate that IMTC significantly improves the diversity of learned options without extrinsic rewards, combined with intrinsic rewards. Moreover, we test the reusability of learned options by transferring options into various tasks, confirming that IMTC helps quick adaptation, especially in complex domains where an agent needs to manipulate objects.

## 1 Introduction

Behavior learning from environmental interaction is a fundamental problem in artificial intelligence and robotics. Recently, combined with deep neural networks (DNN), reinforcement learning (RL) has successfully learned complex behaviors guided by human-designed reward functions (Heess et al., 2017; OpenAI et al., 2019). To reuse trained agents across multiple reward functions, or shortly *tasks*, abstracting a course of action as a higher-level building block (Barto and Mahadevan, 2003; Barto et al., 2013) would be useful. A representative formulation of action abstraction in RL is the options framework (Sutton et al., 1999), where each option consists of a sub policy and its termination condition. Serving as reusable behavioral blocks, options have the potential of accelerating learning in new tasks (Brunskill and Li, 2014). For example, if a wheeled robot already had learned to lean and turn left and right, it can learn to drive on unfamiliar roads faster than learning from scratch. However, discovering reusable options remains challenging due to the difficulty of defining and measuring reusability a priori. Although we can measure the reusability of options when we know the task distribution (e.g., in a Bayesian sense (Solway et al., 2014)), it is difficult to measure it without prior knowledge. Hence, in such cases, we need a reasonable heuristic assumption about future tasks.

In this paper, we follow a common assumption that diversifying options improves the reusability of options (e.g., Barto et al. (2004)). That is, we assume that tasks are uniformly distributed over the state space. Diverse options are expected to work to some extent for *any* task, thus we argue that this is a reasonable heuristics. Under this assumption, the next problem is to measure the diversity of options. The most intuitive strategy would be visiting all states as equally as possible. However, this can be computationally too expensive when the environment has continuous or large state spaces. As an alternative, in the context of learning skills without termination conditions, mutual information (MI) maximization has been widely applied for learning diverse policies (Eysenbach et al., 2019; Baumli et al., 2021; Choi et al., 2021). An advantage of MI maximization is the existence of tractable approximations with DNN, scaling it up to more complex domains.

For learning diverse options, we maximize MI between options and their terminating states conditioned by a starting state. This MI maximization diversifies the destinations of an agent when choosing different options at a state while keeping the transition as deterministic as possible, leading to diverse and meaningful options. Specifically, we propose to maximize this MI by optimizing

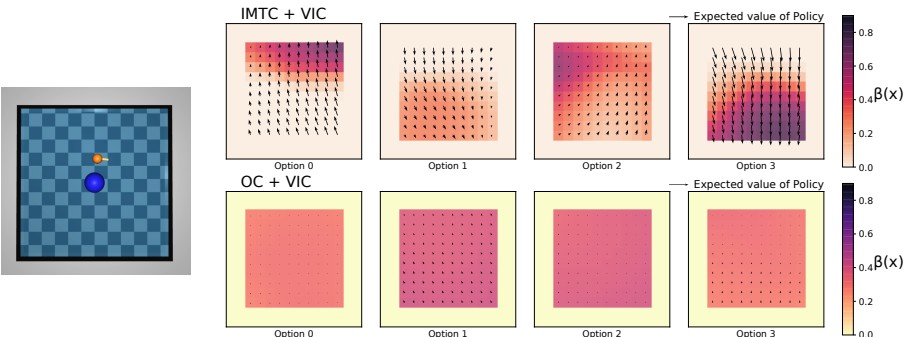

Figure 1: **Left:** `PointBilliard` environment. **Right:** Learned options by IMTC (upper row) and OC (lower row). Arrows show policies, and heatmaps show termination conditions of options.

the termination conditions of options. We derive an unbiased gradient estimator of this MI w.r.t. termination conditions using the termination gradient theorem (Harutyunyan et al., 2019). As a key contribution, we reformulate the estimated gradient using Bayes' rule and derive a tractable approximation with a leaned classifier, yielding the InfoMax Termination Critic (IMTC) algorithm. The easiness of approximation is advantageous compared to the Termination Critic (TC, Harutyunyan et al. (2019)). The same objective is used by Variational Intrinsic Control (VIC, Gregor et al. (2016)) for diversifying sub-policies, but the original VIC used a constant termination probability for all states and options. However, since VIC depends on relating starting and terminating states to an option, we argue that diversifying termination conditions is also important. We show that our method helps VIC learn clearer options.

In this paper, we first introduce backgrounds and notations, followed by the derivation of IMTC. We implement IMTC on the Option Critic Architecture (OC, Bacon et al. (2017)) and Proximal Policy Optimization (PPO, Schulman et al. (2017)). As a minor contribution, our implementation includes a newly proposed optimistic advantage estimation that speeds up option learning for a single task. In experiments, we qualitatively demonstrate that IMTC successfully learns diverse and meaningful options in a reward-free RL (Jin et al., 2020) setting. Figure 1 shows the learned options by IMTC and OC combined with VIC rewards in `PointBilliard` domain. We can see that options learned by IMTC are clearly separated and different. Then we test the reusability of learned options in task adaptation experiments: we first pre-train options without rewards and then transfer them on various tasks. We show that IMTC helps quick adaptation to specific tasks especially in complex tasks where object manipulation is required. Since IMTC is only for learning terminating conditions of options, we can combine it with other methods for learning diverse sub-policies. Our experiments observed that IMTC improves the performance of VIC and RVIC (Baumli et al., 2021).

## 2   BACKGROUND AND NOTATION

We assume the standard RL setting in the Markov decision process (MDP), following Sutton and Barto (2018). MDP $\mathcal{M}$ consists of a tuple $(\mathcal{X}, \mathcal{A}, p, r)$, where $\mathcal{X}$ is the finite set of states, $\mathcal{A}$ is the finite set of actions, $p : \mathcal{X} \times \mathcal{A} \times X \to [0, 1]$ is the state transition function, $r : \mathcal{X} \times \mathcal{A} \to [r_{\min}, r_{\max}]$ is the reward function, where $r_{\min}, r_{\max}$ are minimum and maximum. We use $X_t$ and $A_t$ for denoting random variables of the state and action experienced at time $t$. We define $G \overset{\text{def}}{=} \sum_{t=0}^{\infty} \gamma^t R_t$ as discounted cumulative return, where $R_t = r(X_t, A_t)$ is the reward received at time $t$ and $0 < \gamma < 1$ is the discount factor. We consider maximizing expected discounted return $\mathbb{E}[G|\pi]$. As an agent, we consider a memoryless policy $\pi : \mathcal{X} \times \mathcal{A} \to [0, 1]$. $\pi$ induces two value functions: action-value function $Q^\pi(x, a) \overset{\text{def}}{=} \mathbb{E}_\pi[G|X_0 = x, A_0 = a, \pi]$ and state-value function $V^\pi(x) \overset{\text{def}}{=} \sum_a \pi(a|x)Q^\pi(x, a)$. We use $d^\pi$ for denoting an agent's average occupancy measure $d^\pi(x) = \lim_{n \to \infty} \frac{\mathbb{E}\left[\sum_{t=0}^n \mathbb{I}_{X_t=x}\right]}{n}$, where $\mathbb{I}$ is the indicator function.

Assuming that $\pi$ is differentiable by the policy parameters $\theta_\pi$, the policy gradient (PG) method (Williams, 1992) maximizes $G^\pi$ by updating $\theta_\pi$ via gradient ascent. A common formulation of PG estimates the gradient by $\nabla_{\theta_\pi} G^\pi = \mathbb{E}_{x, a, \pi}\left[\nabla_{\theta_\pi} \log \pi(a|x)\hat{A}(x, a)\right]$, where $\hat{A}(x, a)$ is an estimation of the advantage function $A^\pi(x, a) \overset{\text{def}}{=} Q^\pi(x, a) - V^\pi(x)$. Among many variants of PG

methods, we implemented our method on PPO (Schulman et al., 2017). At each gradient step, PPO updates $\theta_\pi$ to maximize clip$(\frac{\pi(a|x)}{\pi_{\text{old}}(a|x)}\hat{A}, -\epsilon, \epsilon)$, where clip$(x, -\epsilon, \epsilon) = \max(-\epsilon, \min(\epsilon, x))$ and $\pi_{\text{old}}$ is the policy before being updated. This clipping heuristics prevents $\pi$ from updating too rapidly.

**Options Framework**  Options (Sutton et al., 1999) provide a framework for formulating temporally abstracted actions in RL. An option $o \in \mathcal{O}$ consists of a tuple $(\mathcal{I}^o, \beta^o, \pi^o)$, where $\mathcal{I}^o \subseteq \mathcal{X}$ is the initiation $\beta^o : \mathcal{X} \to [0, 1]$ is a termination function, and $\pi^o$ is an *intra-option* policy. Following related studies (Bacon et al., 2017; Harutyunyan et al., 2019), we assume that $\mathcal{I}^o = \mathcal{X}$ focuses on learning $\beta^o$ and $\pi^o$. We let $\mu : \mathcal{X} \times \mathcal{O} \to [0, 1]$ denote a policy over options. A typical RL agent sample $A_t$ from $\pi(\cdot|X_t)$. Analogously, at time $t$, an RL agent with options first samples a termination $T_t$ from $\beta^{O_t}(X_t)$. If $T_t = 1$, $O_{t+1}$ is sampled from $\mu(\cdot|X_t)$ and if not, the current option remains the same. The next action $A_t$ is sampled from $\pi^{O_{t+1}}(\cdot|X_t)$. For option learning methods, we use $\pi$ to denote the resulting policy induced by $\mu$, $\cup_{o \in \mathcal{O}}\pi^o$, and $\cup_{o \in \mathcal{O}}\beta^o$.

**Option value functions**  With options, we have three option-value functions $Q_\mathcal{O}$, $V_\mathcal{O}$, and $U_\mathcal{O}$. $Q_\mathcal{O}$ is the option-value function denoting the value of selecting an option $o$ at state $x$ defined by $Q_\mathcal{O}(x, o) \overset{\text{def}}{=} \mathbb{E}[G|X_0 = x, O_0 = o]$. Similar to the relationship between $Q^\pi$ and $V^\pi$, we let $V_\mathcal{O}$ denote the marginalized option-value function $V_\mathcal{O}(x) \overset{\text{def}}{=} \sum_o \mu(o|x)Q_\mathcal{O}(x, o)$. $U_\mathcal{O}(x, o) \overset{\text{def}}{=} (1 - \beta^o(x))Q_\mathcal{O}(x, o) + \beta^o(x)V_\mathcal{O}(x)$ is called the option-value function *upon arrival* (Sutton et al., 1999) and denotes the value of reaching a state $x$ with $o$ and not having selected the new option. We use these notations in Section 4.2.

**Termination gradient theorem**  Analogously to $p$, we let $P^o : \mathcal{X} \times \mathcal{O} \times X \to [0, 1]$ denote the state transition probability induced by options. When an agent is at $x_s$ and having an option $o$, the probability that $o$ ends at $x_f$ is given by:

$$P^o(x_f|x_s) = \beta^o(x_f)\mathbb{I}_{x_f=x_s} + (1 - \beta^o(x_s))\sum_x p^{\pi^o}(x|x_s)P^o(x_f|x), \tag{1}$$

where $p^{\pi^o}$ is the *policy-induced* transition function $p^{\pi^o}(x'|x) \overset{\text{def}}{=} \sum_{a \in \mathcal{A}} \pi^o(a|x)P(x'|x, a)$. Here we assume that all options eventually terminate, such that $P^o$ is a valid probability distribution over $x_f$. Interestingly, $P^o$ is differentiable with respect to the parameter of $\beta^o$. Harutyunyan et al. (2019) introduced the termination gradient theorem:

**Theorem 1.** *Let $\beta^o$ be parameterized by $\theta_\beta$, and let $\ell_{\beta^o}$ denote the logit of $\beta^o$, i.e., $\ell_{\beta^o} = \log(\frac{\beta^o(x)}{1-\beta^o(x)})$. We have*

$$\nabla_{\theta_\beta}P^o(x_f|x_s) = \sum_x P^o(x|x_s)\nabla_{\theta_\beta}\ell_{\beta^o}(x)(\mathbb{I}_{x_f=x} - P^o(x_f|x)). \tag{2}$$

We use this theorem to derive an unbiased estimator of our target gradient.

## 3  INFOMAX TERMINATION CRITIC

We now present the InfoMax Termination Critic (IMTC) algorithm. To learn diverse options, we propose to maximize the following MI at each state $x_s$:

$$I(X_f; O|x_s) = H(X_f|x_s) - H(X_f|x_s, O) = H(O|x_s) - H(O|X_f, x_s), \tag{3}$$

where $I$ denotes the MI $I(A; B|c) = H(A|c) - H(A|B, c)$, $H$ denotes entropy, and $O$ is a random variable denoting options experienced by an agent. Precisely, we let $\eta$ denote the probability of having an option $o$ when leaving a state $x$: $\eta(o|x) = \beta^o(x)\mu(o|x) + (1 - \beta^o(x))\sum_{x' \in \mathcal{X}} p^{\pi^o}(x|x')\eta(o|x')$ and let $\eta^\pi$ denote $\eta$ marginalized over $d_\pi$: $\eta^\pi(o) = \sum_{x \in \mathcal{X}} d_\pi(x)\eta(o|x)$. Then, we define $O$ as a random variable with $\eta^\pi$. By decomposing this MI as $I(X_f; O|x_s) = H(X_f|x_s) - H(X_f|x_s, O)$, we can interpret this MI maximization as maximization of $H(X_f|x_s)$ and minimization of $H(X_f|x_s, O)$. Maximizing $H(X_f|x_s)$ diversifies possible destinations of an agent. Thus, the resulting options are expected to be diverse in the sense that they likely lead to different destinations. On the other hand, minimizing $H(X_f|x_s, O)$ makes option state transitions more deterministic, leading to more meaningful options.

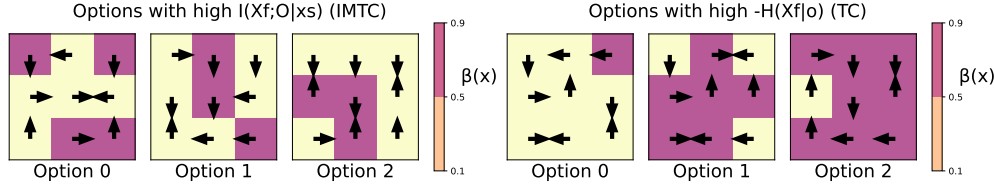

Figure 2: IMTC options (left) and TC options (right).

To build intuition, we searched options that maximize the MI (3) by a simple brute-force search from a limited set of $\pi^o$ and $\beta^o$ in a $3 \times 3$ Gridworld environment. Specifically, we considered only deterministic intra-option polciies and 0.1 or 0.8 as the value of $\beta^o(x)$. The number of options is fixed to $|\mathcal{O}| = 3$. In this environment, an agent can take four actions: Up, Down, Left, and Right. The taken action fails with probability 0.1, and it takes a uniformly random action. Figure 2 shows the result. Arrows represent intra-option policies and colors of the room represent termination probabilities. Here, we can see the tendency that IMTC prefers options that have almost separated termination regions and different policies per option. We also visualized TC (Harutyunyan et al., 2019) options that maximize $-H(X_f|o)$. For TC options, we can see a similar tendency as IMTC options, but options 0 and 1 of TC share the intra-option policies on the left side of the room. Thanks to the diversity term $H(X_f|x_s)$, IMTC successfully avoids such a policy overlap. In addition, we can see that IMTC has smaller termination regions than TC, which can enhance long-range abstraction.

We propose to maximize the MI (3) by updating $\beta^o$ via gradient ascent w.r.t. $\theta_\beta$. For this purpose, we now derive an unbiased estimator of the gradient. First, we write the gradient of the MI using the option transition model $P^o$ and marginalized option-transition model $P(x_f|x_s) = \sum_o \eta^\pi(o) P^o(x_f|x_s)$.

**Proposition 1.** *Let $\beta^o$ be parameterized using a sigmoid function. Given a trajectory $\tau = x_s, \ldots, x, \ldots, x_f$ sampled by $\pi^o$ and $\beta^o$, we can obtain unbiased estimations of $\nabla_{\theta_\beta} H(X_f|x_s)$ and $\nabla_{\theta_\beta} H(X_f|x_s, O)$ by*

$$\nabla_{\theta_\beta} H(X_f|x_s) = \mathbb{E}_{d^\pi, \eta} \left[ -\nabla_{\theta_\beta} \ell_{\beta^o}(x) \beta^o(x) \Big( \log P(x|x_s) - \log P(x_f|x_s) \Big) \right] \tag{4}$$

$$\nabla_{\theta_\beta} H(X_f|x_s, O) = \mathbb{E}_{d^\pi, \eta} \left[ -\nabla_{\theta_\beta} \ell_{\beta^o}(x) \beta^o(x) \Big( \log P^o(x|x_s) - \log P^o(x_f|x_s) \Big) \right], \tag{5}$$

*where $\ell_{\beta^o}(x)$ denotes the logit of $\beta^o(x)$.*

Note that the additional term $\beta^o$ is necessary because $x$ is not actually a terminating state. The proof is based on Section 4 in Harutyunyan et al. (2019) and is given in Appendix A.1.

Now the targetting gradient can be written as:

$$\nabla_{\theta_\beta} I(X_f; O|x_s) = \nabla_{\theta_\beta} H(X_f|x_s) - \nabla_{\theta_\beta} H(X_f|x_s, O)$$
$$= \mathbb{E}_{d^\pi, \eta} \left[ -\nabla_{\theta_\beta} \ell_{\beta^o}(x) \beta^o(x) \left( \log P(x|x_s) - \log P(x_f|x_s) - (\log P^o(x|x_s) - \log P^o(x_f|x_s)) \right) \right], \tag{6}$$

which means that we can approximate MI maximization by estimating $P^o$ and $P$. However, in RL, learning a density model over the state space in RL is often difficult, especially with large or continuous state spaces. For example, it has been tackled with data compression methods (Bellemare et al., 2016) and generative models (Ostrovski et al., 2017). Hence, we reformulate the gradient using Bayes' rule to avoid estimating $P^o$ and $P$. The resulting term consists of the inverse option transition function $p_{\mathcal{O}}(o|x_s, x_f)$, which denotes the probability of having an $o$ given a state transition $x_s, x_f$.

**Lemma 1.** *We now have*

$$\nabla_{\theta_\beta} I(X_f; O|x_s) = \nabla_{\theta_\beta} H(X_f|x_s) - \nabla_{\theta_\beta} H(X_f|x_s, O)$$
$$= \mathbb{E}_{d^\pi, \eta} \left[ \nabla_{\theta_\beta} \ell_{\beta^o}(x) \beta^o(x) \Big( \log p_{\mathcal{O}}(o|x_s, x) - \log p_{\mathcal{O}}(o|x_s, x_f) \Big) \right]. \tag{7}$$

The proof is provided in Appendix A.2. Equation (7) requires estimating of $p_{\mathcal{O}}$ per updating $\pi^o$ and $\beta^o$, which is computational quite expensive. Thus, we approximate approximate the gradient (7) by regressing a classification model over options $\hat{p_{\mathcal{O}}}(o|x_s, x_f)$ on sampled option transition.

## 4 IMPLEMENTATION

Since IMTC can be combined with any on-policy RL methods, we choose PPO (Schulman et al., 2017) as a base algorithm because of its stability and ease of implementation. As notable implementation details, this section explains the estimation of $p_{\mathcal{O}}$ and advantage estimation. We provide further details and the full description of the algorithm in Appendix B.

### 4.1 ESTIMATING $p_{\mathcal{O}}$

To estimate $p_{\mathcal{O}}$, we employ a classification model over options $\hat{p_{\mathcal{O}}}(o|x_s, x_f)$ and regress it on sampled option transitions, as per Gregor et al. (2016). However, our preliminary experiments observed that this online regression could be unstable because the supply of transition data depends on the termination probability and can drastically increase or decrease during training. To address this problem, we maintain a replay buffer $B_{\mathcal{O}}$, which stores option state transitions $\{(o, x_s, x_f)\}$ to stabilize the regression of $\hat{p_{\mathcal{O}}}$. Note that using older option state transitions can introduce bias to $\hat{p_{\mathcal{O}}}$ because it depends on the current policy. However, we found that this is not harmful when the capacity of the replay buffer is reasonably small.

### 4.2 ADVANTAGE ESTIMATION

The original PPO implementation employs GAE (Schulman et al., 2015b) for estimating the advantage, which is important for learning performance (Andrychowicz et al., 2021). Therefore, we employed two variants of GAE in experiments for option learning according to the experimental setup. In the following paragraphs, we let $N$ denote the rollout length used for advantage estimation and let $t + k$ denote the time step at which the current option $o_t$ terminates. Thus, we need to consider the effect of option-switching in advantage estimation When $k < N$. Furthermore, we use two variants of the option-specific TD errors $\delta(o_t) = R_t + \gamma Q_{\mathcal{O}}(x_{t+1}, o_t) - Q_{\mathcal{O}}(x_t, o_t)$ and $\delta_{\mathrm{U}}(o_t) = R_t + \gamma U_{\mathcal{O}}(x_{t+1}, o_t) - Q_{\mathcal{O}}(x_t, o_t)$.

**Independent GAE for Reward-Free RL**   For no-reward experiments with VIC, we used the following variant of GAE:

$$\hat{A}^o_{\mathrm{ind}} = -Q_{\mathcal{O}}(x_t, o_t) + \sum_{i=0}^{\min(k,N)} (\gamma\lambda)^i \delta(o_{t+i}) \tag{8}$$

Here, we ignore the future rewards produced by other options after the current option $o_t$ terminates. This formulation enhances learning diverse intra-option policies per option.

**Upgoing GAE for task adaptation**   For single-task learning, increasing the rollout step $N$ often speeds up learning (Sutton and Barto, 2018). However, future rewards after option termination heavily depend on the selected option and have high variance, especially when learning diverse options. This high variance of future rewards slows advantage learning and causes underestimation of $\hat{A}^o$. Thus, to prevent underestimation, we introduce an *upgoing* GAE (UGAE) for estimating advantage with options:

$$\hat{A}^o_{\mathrm{upg}} = -Q_{\mathcal{O}}(x_t, o_t) + \begin{cases} \sum_{i=0}^{k}(\gamma\lambda)^i \delta^o_{t+i} + \underbrace{\max\left(\sum_{i=k+1}^{N}(\gamma\lambda)^i\delta(o_{t+i}), 0\right)}_{\text{upgoing estimation}} & (k < N) \\ \sum_{i=0}^{N-1}(\gamma\lambda)^i\delta^o_{t+i} + (\gamma\lambda)^N\delta_{\mathrm{U}}(o_{t+N}) & (\text{otherwise}). \end{cases} \tag{9}$$

Like the upgoing policy update (Vinyals et al., 2019), the idea is optimistic regarding future rewards after option termination by taking the maximum over 0. We use $\hat{A}^o_{\mathrm{upg}}$ for task adaptation experiments in Section 5, and confirmed its effectivity in the ablation study in Appendix C.6.

## 5 EXPERIMENTS

As presented in this section, we conducted two series of experiments to analyze the property of IMTC. First, we qualitatively evaluated the diversity of options learned by IMTC with intrinsic rewards, without any extrinsic rewards. Second, we quantitatively test the reusability of learned options by

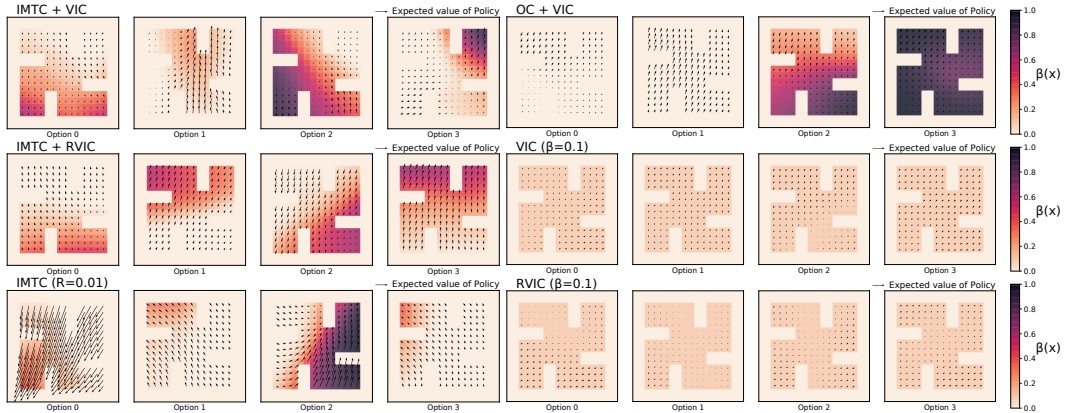

Figure 3: Learned intra-option policies ($\pi^o$) and termination probabilities ($\beta^o$) for each option in PointMaze after training $4 \times 10^6$ steps. Arrows show the expected value of action, and heatmaps show probabilities of each $\beta^o$. **Left:** Options learning by IMTC with three different intrinsic rewards. We can see that intra-option policies have various directions, and termination regions are clearly separated across options. To see the difference between intrinsic rewards, RVIC and constant rewards have an overlapped pair of options (option 1 and 3). Also, we can see that the magnitude of intra-option policies tends larger with constant rewards. **Right:** Options learned by other methods. OC produces a dead option 3 that terminates everywhere and never-ending options 0 and 1. About intra-option policies, all methods successfully avoided learning the same policy, but they only have two directions.

task adaptation on a specific task. As a baseline of termination learning method, we compared our method with OC (Bacon et al., 2017). OC is trained with VIC (Gregor et al., 2016) rewards during pre-training. We did not compare IMTC with TC (Harutyunyan et al., 2019) because our TC implementation failed to learn options with relatively small termination regions as reported in the paper, and there is no official public code for TC[1]. During pre-training without extrinsic rewards, IMTC receives intrinsic rewards when the current option terminates. We compare three IMTC variants with different intrinsic rewards: (i) VIC (Gregor et al., 2016), (ii) RVIC (Baumli et al., 2021), and (iii) constant value ($R_{\text{IMTC}} = 0.01$). Note that $R_{\text{IMTC}} = 0.01$ is chosen from $[0.1, 0.05, 0.01]$ based on the task adaptation results. We also compare IMTC with vanilla VIC and RVIC with fixed termination probabilities. We used $\forall_x \beta^o(x) = 0.1$ since it performed the best in task adaptation experiments, while $0.05$ was used in Gregor et al. (2016). Note that RVIC's objective $I(X_s; O|x_f)$ is different from ours, while IMTC and VIC share almost the same objective. Thus, the use of VIC is more natural, and the combination with RVIC is tested to show the applicability of IMTC. Further details of our VIC and RVIC implementation are found in Appendix B. In order to check only the effect of the different methods for learning beta $\beta^o$, the rest of the implementation is the same for all these methods. That is, OC, vanilla VIC, and vanilla RVIC are also based on PPO and advantage estimation methods in Section 4.2. In this section, we fix the number of options as $|\mathcal{O}| = 4$ for all option-learning methods. We further investigated the effect of the number of options Appendix C, where we confirmed that $|\mathcal{O}| = 4$ is sufficient for most domains. All environments that we used for experiments are implemented on the MuJoCo (Todorov et al., 2012) physics simulator. We further describe the details in Appendix C.

**Option Learning From Intrinsic Rewards** We now qualitatively compare the options learned by IMTC with options of other methods. Learned options depend on the reward structure in the environment, which enables manually designing good reward functions for learning diverse options. Thus, we employed a reward-free RL setting where no reward is given to agents. Instead, each compared method uses some intrinsic rewards, as explained. We fix $\mu$ as $\mu(o|x) = \frac{1}{|\mathcal{O}|}$ in this experiment, since we assume that the future tasks are uniformly distributed. Intra-option policies are trained by PPO (Schulman et al., 2017) and independent GAE (8). We show network architectures and hyperparameters in Appendix C. We set the episode length to $1 \times 10^4$, i.e., an agent is reset to its starting position after $1 \times 10^4$ steps. For all visualizations, we chose the best one from five independent runs with different random seeds.

---

[1]Note that we also could not find any unofficial open source implementation.

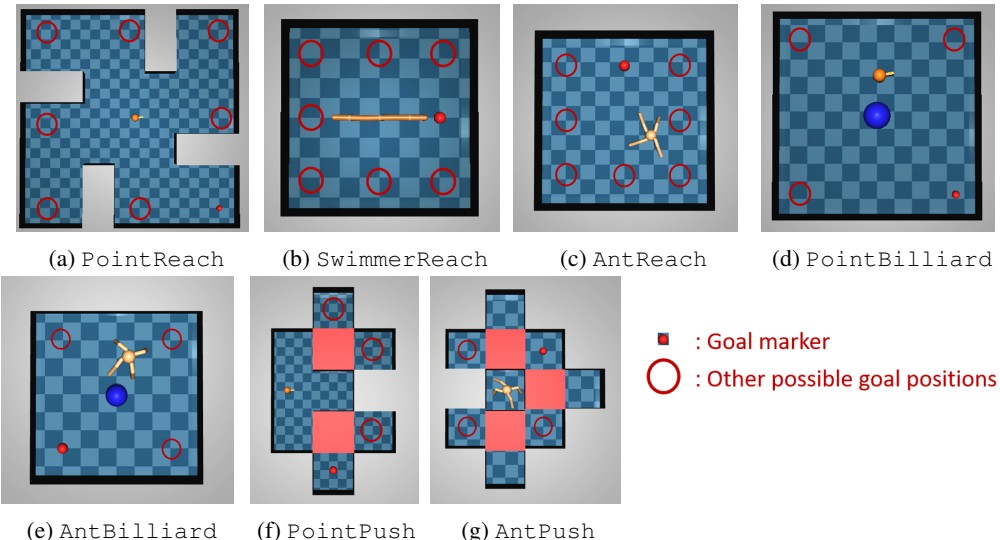

(a) `PointReach`    (b) `SwimmerReach`    (c) `AntReach`    (d) `PointBilliard`

(e) `AntBilliard`    (f) `PointPush`    (g) `AntPush`

Figure 4: Domains used in task adaptation experiments.

We visualized learned options in `PointReach` environment shown in Figure 4a. In this environment, an agent controls the ball initially placed at the center of the room. The state space consists of positions $(\mathbf{x}, \mathbf{y})$ and velocities $(\Delta \mathbf{x}, \Delta \mathbf{y})$ of an agent, and the action space consists of accerations $(\frac{\Delta \mathbf{x}}{\Delta \mathbf{t}}, \frac{\Delta \mathbf{y}}{\Delta \mathbf{t}})$. Figure 3 shows the options learned in this environment after $4 \times 10^6$ steps. Each arrow represents the mean value of intra-option policies, and the heatmaps represent $\beta^o$. In this experiment, we observed the effect of IMTC clearly, for both termination regions and intra-option policies. Interestingly, we don't see clear differences between options learned with VIC and RVIC rewards, while constant rewards tend to make options peaker. OC failed to learn meaningful termination regions: option $0$ and $1$ never terminate, and option $3$ terminates almost everywhere. This result confirms that IMTC can certainly diversify options. Moreover, compared to vanilla VIC and RVIC, intra-option policies learned by IMTC with VIC or RVIC rewards are clearer, in terms of both the magnitude and directions of policies. We believe that this is because diversifying termination regions gives more biased samples to the option classifiers employed by VIC and RVIC. Figure 1 also show options learned by IMTC and OC in `PointBilliard` domain, where we can see the same tendency.

**Transferring skills via task adaptation**  Now we quantitatively test the reusability of learned options by task adaptation with specific reward functions. Specifically, we first trained agents with intrinsic rewards as per the previous section. Then we transferred agents to an environment with the same state and action space but with external rewards. We prepared multiple reward functions, which we call *tasks*, for each domain and evaluated the averaged performance over tasks. We compare IMTC with OC, vanilla VIC, vanilla RVIC, and PPO without pre-training. Also, we compare three variants of IMTC with different intrinsic rewards during pre-training. For a fair comparison, UGAE (9) and PPO are used for all options learning methods. Note that we found UGAE is very effective in this experiments, as we show the ablation study in Appendix C.6. For vanilla VIC and vanilla RVIC, termination probability is fixed to $0.1$ through pre-training and task adaptation. $\epsilon$-greedy based on $Q_{\mathcal{O}}$ with $\epsilon = 0.1$ is used as the option selection policy $\mu$. We hypothesize that diverse options learned by IMTC can help quickly adapt to given tasks, supposing the diversity of tasks.

Figure 4 shows all domains used for task adaptation experiments. For simplicity, all tasks have goal-based sparse reward functions. I.e., an agent receives $R_t = 1.0$ when it satisfies a goal condition, and otherwise the control cost $-0.0001$ is given. Red circles show possible goal locations for each task. When the agent fails to reach the goal after 1000 steps, it is reset to a starting position. `PointReach`, `SwimmerReach`, and `AntReach` are simple navigation tasks where an agent aim to just navigate itself to the goal. We also prepared tasks with object manipulation: in `PointBilliard` and `AntBilliard` an agent aims to kick the blue ball to the goal position, and in `PointPush` and `AntPush`, it has to push the block out of the way to the goal. We pre-traine options learning agents for $4 \times 10^6$ environmental steps and additionally trained them for $1 \times 10^6$ steps for each task. Figure 5 shows learning curves and scatter plots drawn from five independent runs with

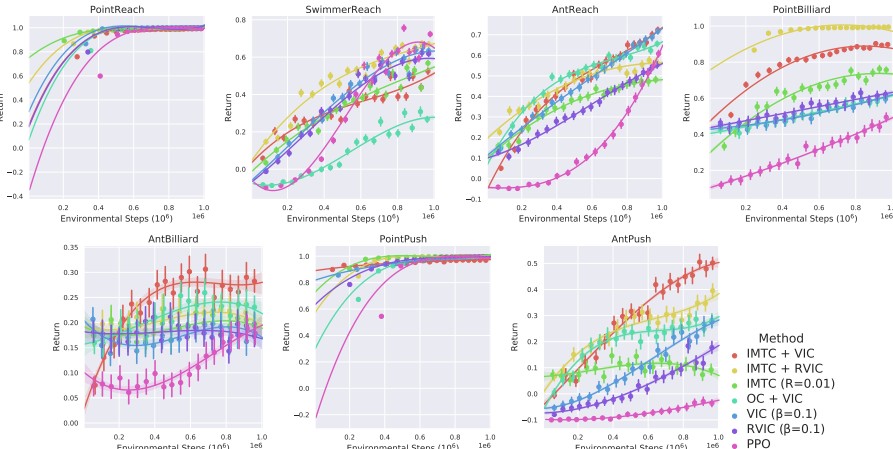

Figure 5: Learning curves for transfer learning experiments.

different random seeds per domain. [2] Here, we observed that IMTC with VIC or RVIC rewards performed the best or was compatible with baselines. IMTC with VIC performed better than OC with VIC except for `AntRearch`, which backs up the effectiveness of diversifying termination regions for learning reusable options. Also, IMTC with VIC and IMTC with RCIC respectively performed better in most of the tasks than VIC and RVIC with fixed termination probabilities. This result suggests that IMTC can boost the performance of option learning methods based on option classifiers, even when the objective is different as with RVIC. On the other hand, IMTC with constant rewards ($R_{\mathrm{IMTC}} = 0.01$) performed worse than IMTC with VIC or RVIC rewards, although it also learned diverse options as we show in Figure 3, suggesting the importance of adjusting rewards. We further analyzed the evolution of intrinsic rewards of VIC and RVIC in Appendix C.5. In addition, we can observe that IMTC's performance is especially better than other methods in relatively complex `PointBilliard`, `AntBilliard`, and `AntPush`, where object manipulation is required. Considering that manipulated balls and boxes move faster than agents in these domains, a choice of options can lead to larger differences in the future state. IMTC is suitable to these domains since it maximizes the diversity of the resulting states, while PPO struggles to learn. Contrary, IMTC's performance is close to other methods in `Reach` tasks, where the goal states are relatively close to the starting states in terms of euclidian distances.

**Gridworld experiments and limitation of the method** Although IMTC successfully learned diverse options in MuJoCo experiments, our analysis in Figure 2 shows the possibility of learning options that are not interesting but have large MI. We further investigated this possibility by visualizing options in a classical four rooms gridworld in Appendix C.8. Interestingly, we observed that IMTC could fall into diverse but unmeaningful options in that environment. We believe that IMTC is often sufficient in a large environment where a randomly-initialized agent rarely gets the same trajectory. However, when the number of possible trajectories is small, diversifying the destinations could be insufficient. In such cases, it can be necessary to extend IMTC to diversify trajectories as done in Sharma et al. (2020).

## 6 RELATED WORK

**Discovering diverse options** Since Sutton et al. (1999) first formulated the options, discovering good options has been challenging. A classic concept for the goodness of options is a bottleneck region (McGovern and Barto, 2001), which refers to important states for reaching diverse areas, such as a passage between two rooms. Once an important region is discovered, we can construct an option that guides an agent to that area. Various approaches have been proposed to define bottleneck regions concisely and compute bottleneck options efficiently, including betweenness (Simsek and Barto, 2008), Eigen options (Machado et al., 2017), successor options (Ramesh et al., 2019), and covering options (Jinnai et al., 2019). Barto et al. (2013) discussed the importance of diverse options for exploration, referring to an option construction method with a graph-based decomposition of an MDP Vigorito and Barto (2010). We share the same motivation with these methods but have a

---

[2]We used seaborn (Waskom, 2021)'s `lmplot` with `order=3` is used to draw learning curves.

different approach. While these methods construct *point options* that bridge two states, we capture a set of states by learning $\beta^o$ directly, making it easy to scale up with function approximation. To our knowledge, (Jinnai et al., 2020) only succeeded in scaling up point options to continuous options using approximated computation of the Laplacian.

**End-to-end learning of options**    As described in the previous paragraph, many studies have attempted to construct options and then train intra-option policies. However, motivated by the recent success of RL with DNN, Bacon et al. (2017) proposed OC to train intra-option policies and termination functions in parallel using a neural network as a function approximator. OC updates $\beta^o$ by gradient ascent, so that maximize $Q_{\mathcal{O}}$. Thus, the resulting terminating regions heavily depend on the reward function. OC also proposed a PG-style method for learning intra-option policies. We used a similar method to OC for learning policies and values, but we employed a different method that does not depend on rewards for learning termination functions. Also, we proposed UGAE (9) for enhancing OC-style policy learning. While OC maximizes the option-value function directly, many heuristic objectives have been proposed with similar architectures, including deliberation cost (Harb et al., 2018), interest (Khetarpal et al., 2020), and safety (Jain et al., 2018). Kamat and Precup (2020) extended OC so that resulting options are diverse by maximizing the divergence between intra-option policies, while IMTC considers the diversity of destinations. Notably, Harutyunyan et al. (2019) proposed the termination critic (TC) that maximizes an information-theoretic objective referred to as *predictability* $-H(X_f|o)$. Our method is inspired by TC and maximizes a similar information-theoretic objective for diversity rather than predictability. In addition, TC requires estimating $P^o(x_f|x_s)$ and a marginal distribution of $P^o$, which can be quite difficult in environments with large or continuous state spaces. We avoid such difficult approximations using Bayes' rule, making IMTC more scalable.

**Mutual Information, Empowerment, and Skill Acquisition**    MI also appears in the literature regarding *intrinsically motivated* RL (Singh et al., 2004), as a driver of goal-directed behavior. A well-known example is the *empowerment* (Klyubin et al., 2005; Salge et al., 2014), obtained by maximizing MI between sequential $k$ actions and the resulting state $I(a_t, ..., a_{t+k}; x_{t+k}|x_t) = H(x_{t+k}|x_t) - H(x_{t+k}|a_t, ..., a_{t+k}, x_t)$. Empowerment represents both large degree of freedom and good preparedness: i.e., larger $H(x_{t+k}|x_t)$ implies that there can be more diverse future states, while smaller $H(x_{t+k}|a_t, ..., a_{t+k}, x_t)$ indicates that an agent can realize its intention with greater certainty. In RL literature, empowerment is often implemented by maximizing the variational lower bounds as intrinsic rewards (Mohamed and Rezende, 2015; Zhao et al., 2020). We can interpret our objective $I(X_f; O|x_s)$ as a variant of empowerment where a fixed number of options represents action sequences. Gregor et al. (2016) also employed this interpretation and introduced VIC for training intra-option policies with fixed termination probabilities in a no reward RL setting. Our experiments observed that IMTC helps VIC to learn meaningful intra-option policies. As a variant of VIC, Baumli et al. (2021) proposed to use the reversed MI $I(X_s; O|x_f)$. MI has been used for discovering diverse skills without termination functions. Eysenbach et al. (2019) proposed maximizing MI between skills and states $I(O; X)$, and Sharma et al. (2020) extended the objective to a conditional MI $I(O; X'|x)$. These methods also maximize variational lower bounds of MI as rewards.

## 7    CONCLUSION

In this paper, we considered the problem of learning diverse options in RL. To learn diverse termination regions of options in a scalable way, we proposed to maximize MI between options and terminating states per starting state. We derived an unbiased gradient estimator to approximately maximize this MI, yielding the InfoMax Termination Critic (IMTC) algorithm. Also, we proposed a practical implementation of IMTC with enhanced advantage estimation in Section 4.2. In reward-free experiments, we visualized that IMTC helped learn diverse and clear options combined with intrinsic rewards. We also showed that options learned by IMTC help an agent to quickly adapt to a specific reward function by transferring learned options. Although our experiments observed that IMTC can learn clear and meaningful options, a potential problem is that learning of a classification model $\hat{p_{\mathcal{O}}}$ heavily depends on exploration. For example, an agent cannot explore a room well, it would not be able to learn sufficiently diverse options. The relationship between diversity and exploration would be interesting. Also, our analysis in Figure 2 and Gridworld experiments in Appendix C.8 suggest that IMTC options can fall into small loops, forming uninteresting options. To prevent this, considering distances between states, e.g., by using bisimulation metric (Castro and Precup, 2010), is a plausible research direction.

## REPRODUCIBILITY STATEMENT

We publish anonymized source code used for all our experiments on https://anonymous.4open.science/r/imtc-anonymized-code-E5D1/.

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

# A  OMITTED PROOFS

## A.1  PROOF OF PROPOSITION 1

First, we make an assumption on the dependence of $\eta$ from $\beta$.

**Assumption 1.** *The empirical distribution of options $\eta(o|x_s)$ is independent of termination conditions* $\cup_{o \in \mathcal{O}} \beta^o$.

This assumption does not strictly hold, and we can consider using, e.g., two-time scale optimization to suppress the distribution shift of $\eta$ by updating $\beta^o$. We employed a replay buffer to mitigate this issue in Section 4.1.

**Lemma 2.** *The following equations hold.*

$$\nabla_{\theta_\beta} H(X_f|x_s) = -\sum_o \eta(o|x_s) \sum_x P^o(x|x_s) \nabla_{\theta_\beta} \ell_{\beta^o}(x)$$
$$\left[ \log P(x|x_s) + 1 - \sum_{x_f} P^o(x_f|x) \Big( \log P(x_f|x_s) + 1 \Big) \right]$$
$$(10)$$

$$\nabla_{\theta_\beta} H(X_f|x_s, O) = -\sum_o \eta(o|x_s) \sum_x P^o(x|x_s) \nabla_{\theta_\beta} \ell_{\beta^o}(x)$$
$$\left[ \log P^o(x|x_s) + 1 - \sum_{x_f} P^o(x_f|x) \Big( \log P^o(x_f|x_s) + 1 \Big) \right]$$
$$(11)$$

Then, sampling $x, x_f$ from $d^\pi$ and $o$ from $\eta$,

$$\nabla_{\theta_\beta} H(X_f|x_s) = \mathbb{E}_{d^\pi, \eta} \left[ -\nabla_{\theta_\beta} \ell_{\beta^o}(x) \beta^o(x) \Big( \log P(x|x_s) - \log P(x_f|x_s) \Big) \right] \quad (12)$$

$$\nabla_{\theta_\beta} H(X_f|x_s, O) = \mathbb{E}_{d^\pi, \eta} \left[ -\nabla_{\theta_\beta} \ell_{\beta^o}(x) \beta^o(x) \Big( \log P^o(x|x_s) - \log P^o(x_f|x_s) \Big) \right]. \quad (13)$$

**Proof of Lemma 2**

*Proof.* First, we prove Equation (10). We have:

$$\nabla_{\theta_\beta} H(X_f|x_s) = -\nabla_{\theta_\beta} \sum_{x_f} P(x_f|x_s) \log P(x_f|x_s)$$

$$= -\sum_{x_f} \left( \nabla_{\theta_\beta} P(x_f|x_s) \log P(x_f|x_s) + P(x_f|x_s) \frac{\nabla_{\theta_\beta} P(x_f|x_s)}{P(x_f|x_s)} \right)$$

$$= -\sum_{x_f} \nabla_{\theta_\beta} P(x_f|x_s) \Big( \log P(x_f|x_s) + 1 \Big)$$

$$= -\sum_{x_f} \sum_o \eta(o|x_s) \underbrace{\nabla_{\theta_\beta} P^o(x_f|x_s)}_{\text{Apply eq. (2)}} \Big( \log P(x_f|x_s) + 1 \Big)$$

$$= -\sum_{x_f} \sum_o \eta(o|x_s) \sum_x P^o(x|x_s) \nabla_{\theta_\beta} \ell_{\beta^o}(x) (\mathbb{I}_{x_f=x} - P^o(x_f|x)) \Big( \log P(x_f|x_s) + 1 \Big)$$

$$= -\sum_o \eta(o|x_s) \sum_x P^o(x|x_s) \nabla_{\theta_\beta} \ell_{\beta^o}(x) \sum_{x_f} (\mathbb{I}_{x_f=x} - P^o(x_f|x)) \Big( \log P(x_f|x_s) + 1 \Big)$$

$$= -\underbrace{\sum_o \eta(o|x_s)}_{\text{sample}} \underbrace{\sum_x P^o(x|x_s)}_{\text{sample}} \nabla_{\theta_\beta} \ell_{\beta^o}(x) \times \left[ \log P(x|x_s) + 1 - \underbrace{\sum_{x_f} P^o(x_f|x)}_{\text{sample}} \Big( \log P(x_f|x_s) + 1 \Big) \right].$$

Sampling $x, x_f, o$, we get (12). Then we prove Equation (11).

$$\nabla_{\theta_\beta} H(X_f|x_s, O) = -\nabla_{\theta_\beta} \sum_o \eta(o|x_s) \sum_{x_f} P^o(x_f|x_s) \log P^o(x_f|x_s)$$

$$= -\sum_o \eta(o|x_s) \sum_{x_f} \left( \nabla_{\theta_\beta} P^o(x_f|x_s) \log P^o(x_f|x_s) + P^o(x_f|x_s) \frac{\nabla_{\theta_\beta} P^o(x_f|x_s)}{P^o(x_f|x_s)} \right)$$

$$= -\sum_o \eta(o|x_s) \sum_{x_f} \underbrace{\nabla_{\theta_\beta} P^o(x_f|x_s)}_{\text{Apply eq. (2)}} \left( \log P^o(x_f|x_s) + 1 \right)$$

$$= -\sum_o \eta(o|x_s) \sum_{x_f} \sum_x P^o(x|x_s) \nabla_{\theta_\beta} \ell_{\beta^o}(x) (\mathbb{I}_{x_f=x} - P^o(x_f|x)) \left( \log P^o(x_f|x_s) + 1 \right)$$

$$= -\sum_o \eta(o|x_s) \sum_x P^o(x|x_s) \nabla_{\theta_\beta} \ell_{\beta^o}(x) \sum_{x_f} (\mathbb{I}_{x_f=x} - P^o(x_f|x)) \left( \log P^o(x_f|x_s) + 1 \right)$$

$$= -\underbrace{\sum_o \eta(o|x_s)}_{\text{sample}} \underbrace{\sum_x P^o(x|x_s)}_{\text{sample}} \nabla_{\theta_\beta} \ell_{\beta^o}(x) \times \left[ \log P^o(x|x_s) + 1 - \underbrace{\sum_{x_f} P^o(x_f|x)}_{\text{sample}} \left( \log P^o(x_f|x_s) + 1 \right) \right]$$

Sampling $x, x_f, o$, we get eq. (13). $\qquad\square$

## A.2 PROOF OF LEMMA 1

*Proof.* First, using Bayes' rule, we have:

$$P^o(x_f|x_s) = \frac{\Pr(o|x_f, x_s) \Pr(x_f|x_s)}{\Pr(o|x_s)} = \frac{p_{\mathcal{O}}(o|x_f, x_s) P(x_f|x_s)}{\eta(o|x_s)}$$

Then, we have:

$$\log P^o(x_f|x_s) - \log P(x_f|xs) = \log \frac{P^o(x_f|x_s)}{P(x_f|xs)}$$

$$= \log \frac{\frac{p_{\mathcal{O}}(o|x_s, x_f) P(x_f|x_s)}{\eta(o|x_s)}}{P(x_f|xs)}$$

$$= \log \frac{p_{\mathcal{O}}(o|x_s, x_f)}{\eta(o|x_s)}$$

Applying this equation to the eq. (6), we have:

$$\nabla_{\theta_\beta} I(X_f; O|x_s) = \nabla_{\theta_\beta} H(X_f|x_s) - \nabla_{\theta_\beta} H(X_f|x_s, O)$$

$$= \mathbb{E}_{d^\pi, \eta} \left[ -\nabla_{\theta_\beta} \ell_{\beta^o}(x) \beta^o(x) \left( \log P(x|x_s) - \log P(x_f|x_s) - \log P^o(x|x_s) + \log P^o(x_f|x_s) \right) \right]$$

$$= \mathbb{E}_{d^\pi, \eta} \left[ \nabla_{\theta_\beta} \ell_{\beta^o}(x) \left( \left( \log P^o(x|x_s) - \log P(x|x_s) \right) - \left( \log P^o(x_f|x_s) - \log P(x_f|x_s) \right) \right) \right]$$

$$= \mathbb{E}_{d^\pi, \eta} \left[ \nabla_{\theta_\beta} \ell_{\beta^o}(x) \left( \log \frac{p_{\mathcal{O}}(o|x_s, x)}{\eta(o|x_s)} - \log \frac{p_{\mathcal{O}}(o|x_s, x_f)}{\eta(o|x_s)} \right) \right]$$

$$= \mathbb{E}_{d^\pi, \eta} \left[ \nabla_{\theta_\beta} \ell_{\beta^o}(x) \left( \log p_{\mathcal{O}}(o|x_s, x) - \log p_{\mathcal{O}}(o|x_s, x_f) \right) \right]$$

$\qquad\square$

## B IMPLEMENTATION DETAILS

**Clipped $\beta$ loss** Common PPO implementation updates $\pi_\theta$ multiple times. However, our preliminary experiments observed that performing multiple updates for $\beta^o$ led to destructively large updates and

---

**Algorithm 1** InfoMax Termination Critic with VIC, PPO Style

---

1:  **Given:** Initial option-value $Q_{\mathcal{O}}$, option-policy $\pi^o$, and termination function $\beta^o$.
2:  Let $B_{\mathcal{O}}$ be a replay buffer for storing option-transitions.
3:  **for** $k = 1, ...$ **do**
4:      **for** $i = 1, 2, ..., N$ **do**                                   ▷ Collect experiences from environment
5:          Sample termination variable $b_i$ from $\beta^{o_i}(x_i)$
6:          **if** $b_i = 1$ **then**
7:              Store $(x_s, x_f, o_i), (x_{s+1}, x_f, o_i), ..., (x_{s+h}, x_f, o_i)$ to the replay buffer $B_{\mathcal{O}}$
8:          **end if**
9:          Choose next option $o_{i+1}$ by $\epsilon$-Greedy
10:          Receive reward $R_i$ and state $x_{i+1}$, taking $a_i \sim \pi_{o_{i+1}}(x_i)$
11:      **end for**
12:      **for** $k = 1, 2, ...,$ Num. of PPO epochs **do**                 ▷ Optimize $\pi^o, Q_{\mathcal{O}}$, and $\beta^o$
13:          **for all** $x_i$ in the trajectory **do**
14:              Compute $R_{\text{VIC}}$ from the target network
15:              Compute $\hat{A}^o_{\text{ind}}$ by (8) using $R_{\text{VIC}}$
16:              Update $\pi^o(a_i|x_i)$ via PPO using $\hat{A}^o$
17:              Update $Q_{\mathcal{O}}(x_i, o)$ to regress to $\hat{A}^o$
18:              **if** $o_i$ has already terminated **then**
19:                  Update $\beta^o(x_i)$ via (7)
20:              **end if**
21:          **end for**
22:      **end for**
23:      Train $\hat{p}$ and $\hat{\mu}$ by option-transitions sampled from $B_{\mathcal{O}}$
24:      **if** $k \mod K_{\text{VIC}}$ **then**
25:          Update the target network for VIC
26:      **end if**
27: **end for**

---

resulted in the saturation of $\beta^o$ to zero or one. Hence, to perform PPO-style multiple updates, we introduce a clipped objective of eq. (7):

$$L^{\text{CLIP}}(\theta_\beta) = \text{clip}(\ell_{\beta^o}(x) - \ell_{\beta^o_{\text{old}}}(x), -\epsilon_\beta, \epsilon_\beta)\beta^o_{\text{old}}(x)\Big( \log p_{\mathcal{O}}(o|x_s, x) - \log p_{\mathcal{O}}(o|x_s, x_f) \Big),$$
(14)

where $\epsilon_\beta$ is a small coefficient, and $\beta^o_{\text{old}}$ is an old $\beta^o$ before updating. We also add maximization of the entropy of $\beta^o$ for preventing the termination probability saturating on zero or one. To this end, we maximize $L^{\text{CLIP}}(\theta_\beta) + c_{H_\beta} H(\beta^o(x))$ via gradient ascent w.r.t. $\theta_\beta$, where $c_{H_\beta}$ is a weight of the entropy bonus.

**Full description of the algorithm**    Algorithm 1 shows a full description of our implementation of IMTC on PPO when combined with VIC rewards. As of the original PPO, it is built on the A2C-style (Mnih et al., 2016; Wu et al., 2017) architecture with multiple synchronous actors and a single learner. First, we collect $N$-step experiences interacting with environments. At line 7, we append tuples corresponding to option transitions $(x_s, x_f, o_i), ..., (x_{s+h}, x_f, o_i)$ to $B_{\mathcal{O}}$. Here, we do not use all transitions and store first $h_{\max}$ options to prevent memory shortage. We used $h_{\max} = 10$ or $h_{\max} = 20$. Then we update $\pi^o, Q_{\mathcal{O}}$, and $\beta^o$. Line 13 is done via minibatch sampling in the actual implementation. We also update $\hat{p_{\mathcal{O}}}$ for estimating the gradient (7), sampling from the replay buffer $B_{\mathcal{O}}$. We empirically found that rapidly changing $R_{\text{VIC}}$ leads to unstable learning, especially when IMTC is used in parallel. Thus, we employ a target network to compute $R_{\text{VIC}}$ and periodically update it at line 26. We used $K_{\text{VIC}} = 20$ in the experiments.

**Our implementation of VIC and RVIC**    In experiments, we show that IMTC helps an agent learn diverse options without reward signals. For this purpose, we employ VIC (Gregor et al., 2016) and RVIC (Baumli et al., 2021) for providing intrinsic rewards. Here we explain our VIC and RVIC implementation.    VIC updates intra-option policies to maximize the lower bound of MI (3) $H(O|x_s) - H(O|x_s, X_f) \geq H(O|x_s) + \mathbb{E}_{o,x_f}[\log q(o|x_s, x_f)]$ as rewards,

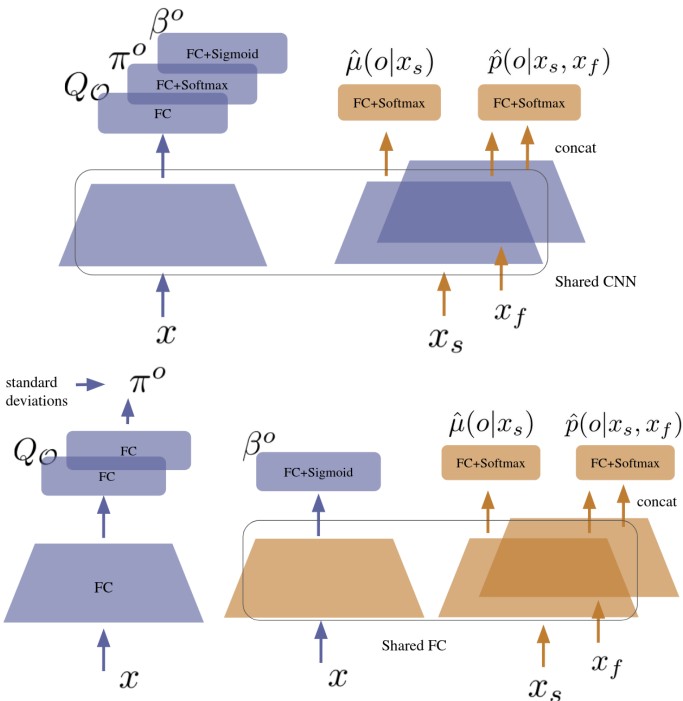

Figure 6: Neural Network architecture used for the Gridworld experiments (top) and continuous control tasks (bottom).

where $q$ can be any distribution and called an *option inference model*. We use $\hat{p_\mathcal{O}}$ as $q$. We also learn $\hat{\eta}(o|xs)$ from sampled option transitions and approximate $H(O|x_s)$ by $H(O|x_s) = -\sum_{o \in \mathcal{O}} \eta(o|x_s) \log \eta(o|x_s) = -\mathbb{E}[\log \eta(o|x_s)] \approx \log \hat{\eta}(o|x_s)$. To this end, we giving an agent $R_{\mathrm{VIC}} = c_{\mathrm{VIC}} (\log \hat{p_\mathcal{O}}(o|x_s, x_f) - \log \hat{\eta}(o|x_s))$ as an immediate reward when an option $o$ terminates, where $c_{\mathrm{VIC}}$ is a scaling coefficient. This is different from the original VIC implementation where $H(O|x_s)$ is treated as a constant. However, we empirically found this formulation helps diversity options in our preliminary experiments. In case of RVIC, we add a another classifier $\hat{q}(o|x_f)$, and constructed rewards by $R_{\mathrm{RVIC}} = c_{\mathrm{VIC}} (\log \hat{p_\mathcal{O}}(o|x_s, x_f) - \log \hat{q}(o|x_f)) \hat{q}$ is learned in the same fashion as $\hat{p_\mathcal{O}}$ and $\hat{\eta}$ using the replay buffer $B_\mathcal{O}$.

## C  EXPERIMENTAL DETAILS

Our anonymized code used for all our experiments is on https://anonymous.4open.science/r/imtc-anonymized-code-E5D1/.

### C.1  NETWORK ARCHITECTURE

Figure 6 illustrates the neural network architecture used in our experiments. We used a shared convolutional encoder for Gridworld experiments, and two separated fully connected layers with 64 units for continuous control experiments. $\pi^o$ is parameterized as a Gaussian distribution with separated networks for standard derivations per option, similar to Schulman et al. (2015a). We used ReLU as an activator for all hidden layers and initialized networks by the orthogonal (Saxe et al., 2014) initialization in all experiments. Note that the last layer for intra-option policy $\pi^o$ was initialized with small values, following the standard practice (Andrychowicz et al., 2021). We used the Adam (Kingma and Ba, 2015) optimizer in all experiments. Unless otherwise noted, we used the default parameters in PyTorch (Paszke et al., 2019) 1.8.1.

| Description | Value |
|---|---|
| $\gamma$ | 0.99 |
| Adam Learning Rate | $3 \times 10^{-4}$ |
| Adam $\epsilon$ | $1 \times 10^{-4}$ |
| Clip parameter for $\ell_{\beta^o}$ ($\epsilon_\beta$) | 0.05 |
| Num. timesteps per rollout | 256 |
| Num. actors | 16 |
| GAE $\lambda$ | 0.95 |
| Num. epochs for PPO | 10 |
| Minibatch size for PPO | 1024 |
| Weight of $H(\pi^o)$ ($c_H$) | 0.001 |
| Weight of $H(\beta^o)$ ($c_{H_\beta}$) | 0.01 |
| Gradient clipping | 0.5 |
| Capacity of $B_{\mathcal{O}}$ | 8192 |
| Max num. option transitions to store ($h_{\max}$) | 20 |
| Num. epochs for training $\hat{p}$ and $\hat{\mu}$ | 4 |
| Minibatch size for training $\hat{p}$ and $\hat{\mu}$ | 2048 |
| Scaling of $R_{\mathrm{VIC}}$ ($c_{\mathrm{VIC}}$) | 0.005 |
| Synchronizing interval of the target network for VIC ($K_{\mathrm{VIC}}$) | 20 |

Table 1: Used hyperparameters

## C.2 HYPERPARAMETERS

Table 1 shows all hyperparameters used in IMTC + VIC experiments on MuJoCo continuous control tasks.

## C.3 ENVIRONMENT IMPLEMENTATION

Gridworld is based on RLPy (Geramifard et al. (2015), BSD3 License). We constructed continuous control environments on the MuJoCo (commercial license, Todorov et al. (2012)), using OpenAI Gym (MIT license, Brockman et al. (2016)). Especially, point environments are implemented based on "PointMaze" in rllab (MIT license, Duan et al. (2016)) with some modifications, mainly around collision detection. We also refered to the modified PointMaze code[3] (Apache 2.0 license) relased by Nachum et al. (2018). The swimmer robot is originally used in Coulom (2002).

## C.4 COMPUTATIONAL RESOURCES

All experiments are conducted on a private cluster with NVIDIA P100 GPUs. On the cluster, training IMTC with VIC on MuJoCo PointMaze domain for $4 \times 10^6$ steps takes about 27 minutes.

## C.5 COMPARISON OF VIC AND RVIC REWARDS

Figure 7 shows the

## C.6 EFFECTIVITY OF UGAE

To investigate the effect of UGAE (9), we compared IMTC and OC with and without UGAE in task adaptation experiments. IMTC (no UGAE) and OC (no UGAE) estimate the adavantage ignoring the future rewards after switching options, similar to advantage estimation proposed by Bacon et al. (2017). Figure 8 shows the result. We can see that UGAE improves the performance in all domains for both IMTC and OC.

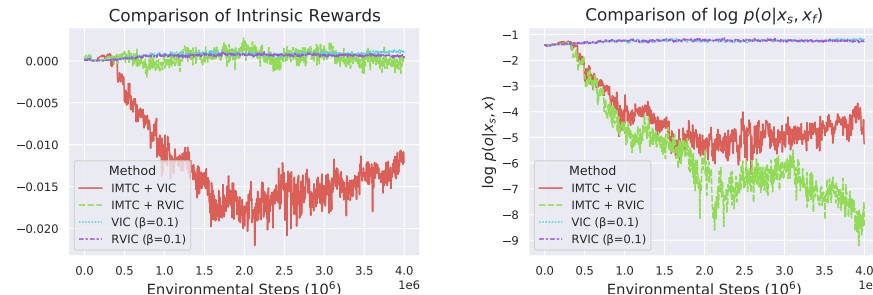

Figure 7: **Left:** Comparison of intrinsic rewards during training. **Right:** Comparison of $\log \hat{p_{\mathcal{O}}}(o|x_s, x)$ during training.

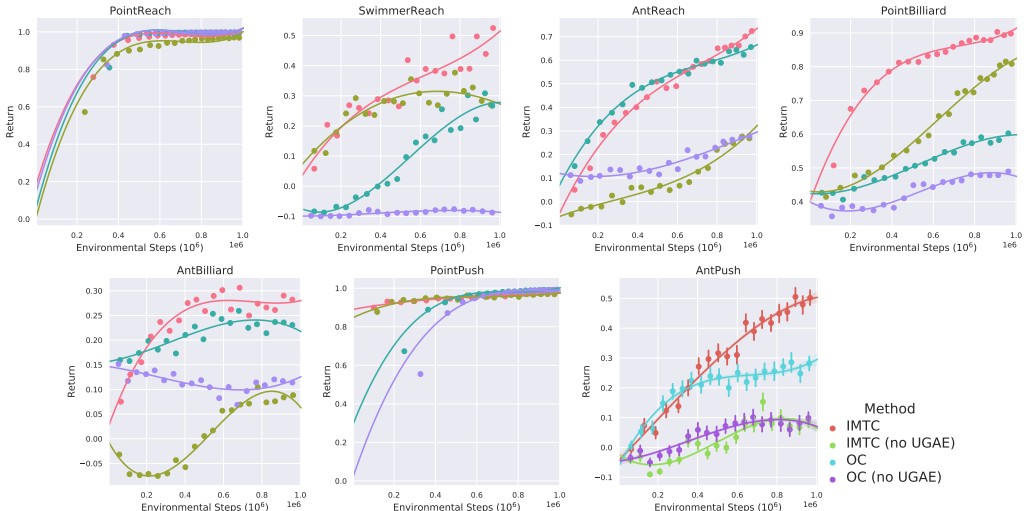

Figure 8: Learning curves for transfer learning experiments with

### C.7 NUMBER OF OPTIONS

Figure 9 shows learning curves of IMTC agents with variable number of options $(2, 4, 6, 8, 10, 12)$ in task adaptation experiments. Increasing the number of options sometimes improves the performance (e.g., in SwimmerReach), but the relationship is not obvious. However, we can confirm that increasing the number of options does not have a bad effect on the performance, while it reduces training steps per each intra-option policy, thanks to network sharing Appendix C.1.

### C.8 OPTIONS LEARNED IN FOUR ROOMS

Figure 10 show the options learned in the classical Four Rooms Gridworld (Sutton et al., 1999).

---

[3] https://github.com/tensorflow/models/tree/v2.3.0/research/efficient-hrl

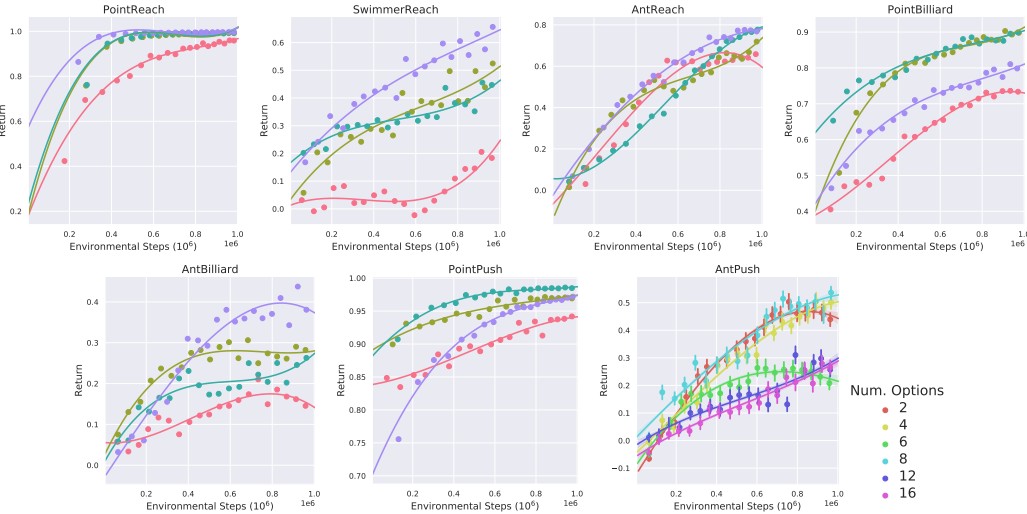

Figure 9: Learning curves for transfer learning experiments.

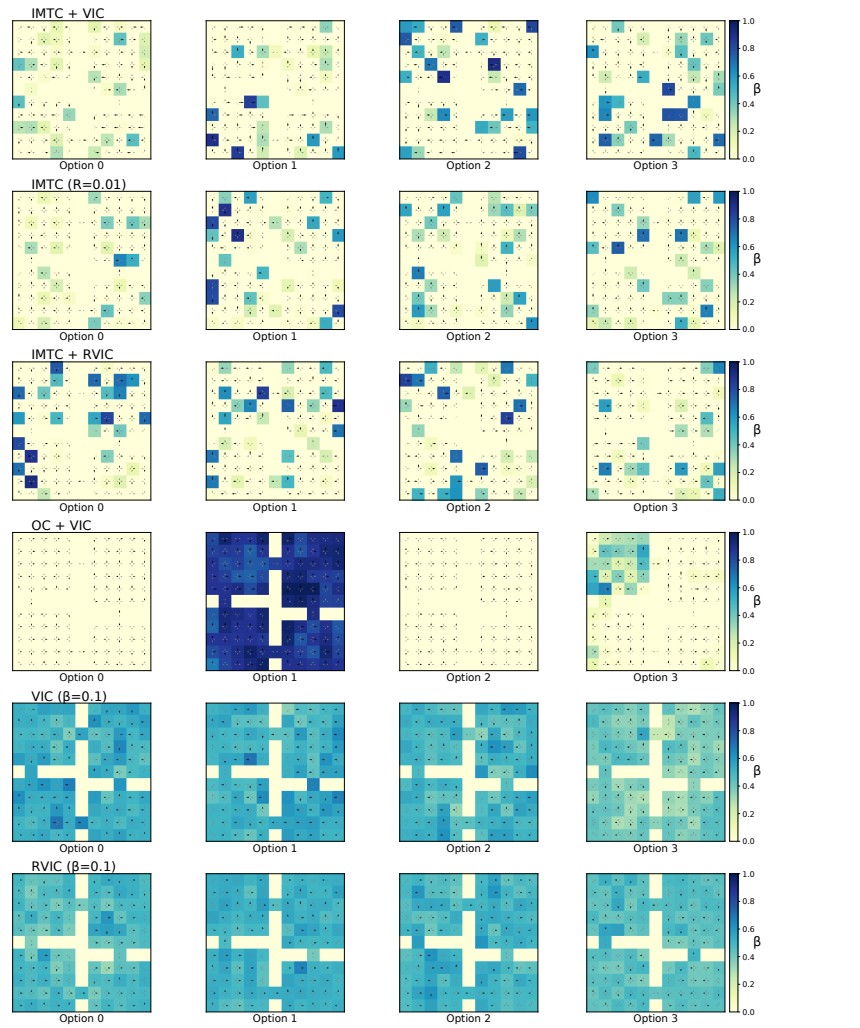

Figure 10: Options learned in four rooms environment

