# OpenReview forum: "Learning Diverse Options via InfoMax Termination Critic"
_ICLR.cc/2022/Conference — ICLR 2022 Submitted_

### Official Review · Reviewer_YG8c · 2021-10-21

**Correctness:** 2
**Technical Novelty And Significance:** 3
**Empirical Novelty And Significance:** 2
**Recommendation:** 6
**Confidence:** 4

**Main Review:**

The general idea of using the termination gradient in VIC is novel and interesting. That said, I feel that the method which was implemented has some gap from the theory, and that the method that is eventually implemented is too similar to VIC. Since the VIC paper was published there were quite a few papers extending it, like DIAYN and RVIC to give two examples. Thus, the novelty of this work is more limited, as it is only comparing to VIC as a baseline. I also feel that I don't learn enough from the experimental section and the algorithm description about the actual difference proposed here.

More concretely, the implementation of the option model is not clear to me and is not well justified. The theoretical derivation suggests using a model that predicts the probability of seeing s_f after executing option o from state s. However in practice, this model is not used, and it is replaced with the discriminator as being done in VIC, which is a different kind of a model. Furthermore, using this model instead of the option model makes the methods much more similar empirically to VIC and much less novel in my opinion. I think this should be clarified and discussed in much more detail, e.g., what is exactly the difference from VIC.

Section 4.2 makes some decisions that are not very clear to me and I would have like to see some experiments supporting them. I don't see it as a major part of the paper though and it might have been better to put this section in the supplementary.

There seems to be a mistake in the definition of the "undiscounted occupancy measure". First of all, it is unclear which occupancy measure is it; is it the average reward occupancy? is it the finite horizon? being undiscounted doesn't tell much. Secondly, on what is the expectation being taken over? if it is on trajectories executed by the policy, then shouldn't the state occupancy be infinity for all non transient states? the average reward and finite horizon state occupancies do not suffer from this issue

**Summary Of The Paper:**

The authors propose an HRL algorithm that uses the VIC objective to discover the options, i.e., the termination condition of the options is trained to maximize the mutual information between the set of options and their terminating states. These options are first trained without access to a reward function in an unsupervised manner. Later on there are experiments that show how an RL agent can re-use these options in downstream tasks.

**Summary Of The Review:**

Nice idea, but the connection between the theory and practice is somewhat missing, resulting in an algorithm that might be too similar to VIC, and the differences are not explained clearly enough nor supported by the experiments. In addition, including more baselines would have made the empirical contributions stronger.

---

> ### Author Response · Authors · 2021-11-23
> **Response to Reviewer YG8c**
>
> Thank you for your thoughtful comments.
> We will clarify the difference from VIC in the revised manuscript.
>
> > The theoretical derivation suggests using a model that predicts the probability of seeing $s_f$ after executing option o from state s. However in practice, this model is not used, and it is replaced with the discriminator as being done in VIC, which is a different kind of a model.
>
> We agree that the initial equation expansion (Proposition 1) suggests the use of option transition model $P^o(x_f|x_s)$.
> However, we then derived that this gradient is also expressed by the option classification model $\hat{p}_\mathcal{O}(o|x_s, x_f)$.
> Here, our intention is to avoid the estimation of $P^o$.
> Our objective $I(X_f;O|x_s)$ depends on the current options, so the learned $P^o$ must quickly adapt to the change of $\pi^o$ and $\beta^o$, which we found difficult.
> In our preliminary experiments, we failed to learn $P^o$ in such an online manner with limited data and found it easier to maintain $\hat{p}_\mathcal{O}$.
> We will clarify this point in the revised paper.
>
> > Furthermore, using this model instead of the option model makes the methods much more similar empirically to VIC and much less novel in my opinion.
>
> We agree that the option classifier used in our method is similar to the one used in VIC.
> However, the derivation and the role of the classifier are very different between the two methods.
> Our method uses this classifier to update $\beta^o(x)$ by gradient ascent.
> However, VIC uses the classifier to approximate a variational lower bound of $I(X_f;O|x_s)$, and update option policies by any reinforcement learning algorithm that uses the approximate lower bound as rewards.
> In short, the derivation of VIC says nothing about updating $\beta^o(x)$.
> Moreover, we believe that diversifying the termination regions of options by our method improves the performance of VIC and similar methods, since it helps $\hat{p}_\mathcal{O}$ to classify option transitions more clearly.
> Our experiments actually observed the improvement of adaptation performance.
> In the revised manuscript, we will note more on the relationship between VIC and our method, especially on why diversifying the termination regions improves VIC.
> Also, we will add a new experiment with IMTC + RVIC to show the wide applicability of IMTC.
>
>
> > Section 4.2 makes some decisions that are not very clear to me and I would have like to see some experiments supporting them. I don't see it as a major part of the paper though and it might have been better to put this section in the supplementary.
>
> The later trick (upgoing GAE) is actually important in the task-adaptation phase as we show in Appendix C, and we believe that this technique is effective for other option learning methods like option-critic.
> We will clarify this point in the revised manuscript.
>
> > There seems to be a mistake in the definition of the "undiscounted occupancy measure".
>
> Thank you for your correction. We will use the averaged occupancy in the revised manuscript.

---

> > ### Comment · Reviewer_YG8c · 2021-11-26
> > **Reply**
> >
> > Thank you for the revised manuscript and addressing my concearns. I have raised my score to 6.

---

> > > ### Author Response · Authors · 2021-11-30
> > > **Thank you for checking the revised manuscript**
> > >
> > > We appreciate your understanding of our paper and your efforts in reviewing it.

---

### Official Review · Reviewer_rEv8 · 2021-10-31

**Correctness:** 3
**Technical Novelty And Significance:** 2
**Empirical Novelty And Significance:** 2
**Recommendation:** 6
**Confidence:** 4

**Main Review:**

Overall I enjoyed the paper, it is well organized and well written, and it touches in a topic that I personally find very interesting which is options learning.

Strengths:
- Well written paper and easy to follow.
- Straight forward objective with a nice intuitive explanation. Should be easy to re-implement for the reader.
- Evaluation done in a set of diverse tasks.

Weakness:
- Maybe a concrete definition of what's meant by diversity in this paper could help. The objective encourages the options to learn different terminating states, but not different trajectories. So, based on the objective two options that "move right" for 38 steps, and one moves "up" on step 39 and the other moves "down" on step 39 would be diverse, and would maximize the objective.
If this is not a behavior that does not happen in practice, it would be nice to have a discussion around it, since nothing strictly prohibits this from happening.

- Where the number of options pre-defined? If so, how did you pick that number?

- On eq. 9, what is the assumption on the reward function to take the maximum over 0 as optimistic. What happens if all rewards are positive in a task? Taking the max over 0 would probably not help with understimation.

- Looking at the results, it's interesting to see that "reach" tasks with large action spaces is where the proposed method tends to underperform. Does the complexity of the task or the action space play a role in how difficult it is to learn options? I would like to see some move in-depth analysis on the results.

- Following on the results, I would suggest avoiding making non-specific statements like: "IMTC outperforms other methods with a large margin in complex PointBilliard, AntBilliard, and AntPush". In that case, my next question is "what is a large margin?" How significant are these values? How many times did you re-run the experiments to make that claim?
I would like to see some more specific numbers around these results.


Nitpick:
- At the beginning of the paper, there is quite a bit of emphasis on "we hypothesis diverse options are reusable". This is a generally well-accepted hypothesis, but the way its written it makes it sound that no one thought of it before.




**Summary Of The Paper:**

The authors propose an objective for learning a diverse set of options in which the goal is to maximize the entropy of terminating states from any initial state while at the same time minimize the entropy of terminating states given a specific option.
Intuitively, this means learning options that tend to be deterministic, while at the same time are able to reach diverse terminating states.
The authors then evaluate against other option learning frameworks in a series of diverse environments.

**Summary Of The Review:**

Well written paper with an interesting objective that makes intuitive sense.
The results look promising, but the discussion and analysis could have more depth; there are questions left unanswered on this paper.

---

> ### Author Response · Authors · 2021-11-23
> **Response to Reviewer rEv8**
>
> Thank you for your detailed and inspiring comments.
> We are now working on an additional empirical analysis of our method.
>
> > Maybe a concrete definition of what's meant by diversity in this paper could help. The objective encourages the options to learn different terminating states, but not different trajectories. So, based on the objective two options that "move right" for 38 steps, and one moves "up" on step 39 and the other moves "down" on step 39 would be diverse, and would maximize the objective. If this is not a behavior that does not happen in practice, it would be nice to have a discussion around it, since nothing strictly prohibits this from happening.
>
> Thank you for raising an interesting question. We will discuss this problem in section 5 and appendix C in the revised manuscript.
> First, we mainly consider the diversity of goals of an agent.
> About not considering the diversity of trajectories, yes, we think that it can be problematic in some cases.
> In our method, an agent updates $\beta_o$ based on the learned classifier $\hat{p}_\mathcal{O}(o|x_s, x_f)$, and $\hat{p}_\mathcal{O}$ is learned from sampled state transitions.
> Thus, if $\hat{p}_\mathcal{O}$ is learned from sample transitions with almost the same trajectories but with different terminating states, the resulting options can reflect it, and the 'fake diversity problem can happen.
> We actually observed such a problem in our Gridworld experiments, where the number of possible trajectories is considerably smaller than in MuJoCo experiments.
> We believe that we did not observe this problem in MuJoCo experiments because a randomly-initialized agent rarely gets the same trajectory in this setting.
>
> > Where the number of options pre-defined? If so, how did you pick that number?
>
> We used four options for all experiments in the main paper, as we briefly explained at the beginning of section 5.
> To decide the number of options, we tested $|\mathcal{O}| = [2, 4, 6, 8]$ in Appendix C.6, and we observed that four options are sufficient in many cases. We will clarify this point in the revised manuscript.
>
> > On eq. 9, what is the assumption on the reward function to take the maximum over 0 as optimistic.
>
> There, we take the maximum of 0 and an estimated advantage. Since advantages should converge to 0 in theory, we called taking the maximum over 0 optimistic.
>
> > Looking at the results, it's interesting to see that "reach" tasks with large action spaces is where the proposed method tends to underperform. Does the complexity of the task or the action space play a role in how difficult it is to learn options? I would like to see some move in-depth analysis on the results.
>
> This is an interesting point.
> We agree that the cardinality of action space would matter. However, we believe that what is more important is the distance to the goal state. IMTC is more advantageous in tasks where goal states are more distant in terms of euclidian distance due to its nature of seeking the diversity of option terminating state. We will add more discussion in the revised manuscript.
>
> > Following on the results, I would suggest avoiding making non-specific statements
>
> Thank you for your suggestion. We will try to make the discussion more explicit in the revised manuscript.
>
> > At the beginning of the paper, there is quite a bit of emphasis on "we hypothesis diverse options are reusable". This is a generally well-accepted hypothesis, but the way its written it makes it sound that no one thought of it before.
>
> Thank you for your suggestion. We will explicitly note that this is a common idea in the revised manuscript.

---

> > ### Comment · Reviewer_rEv8 · 2021-12-03
> > **Thank you for the response**
> >
> > I appreciate taking the time to address my comments. I will adjust my score accordingly.
> > Overall I enjoyed reading this paper, and there are quite a few nice ideas.
> > The one thing that prevented me from giving a higher score is that the definition of diversity feels a bit limiting for the issue I mentioned above, and I still have my doubts whether the current definition is generally useful or would be limiting depending on the scenario.
> > It would be great if follow up work could expand on this definition.

---

> > > ### Author Response · Authors · 2021-12-06
> > > **Thank you for your review and responce.**
> > >
> > > Thank you for checking our revised manuscript. We really appreciate your efforts.
> > >
> > > Here let us discuss your point a bit further.
> > >
> > > >  I still have my doubts whether the current definition is generally useful or would be limiting depending on the scenario. It would be great if follow up work could expand on this definition.
> > >
> > > We agree with your point. We employed 'goal-based diversity' in this paper, but we believe that some other definitions are possible, such as trajectory-based or state occupancy-based diversity. However, especially in environments with continuous state and action spaces, there can be infinitely many trajectories and state-occupancy. So, we believe that it is an interesting topic that what kind of abstractions are effective for capturing meaningful diversity in such scenarios. Our goal-based diversity approach is an instance of heuristic abstractions for this purpose (although it is somewhat limited, as you pointed out!). Also, the computational cost of learning or approximating some measure on those spaces would be interesting. DADS (https://openreview.net/forum?id=HJgLZR4KvH) employed a model-based method for capturing trajectory-based diversity, but we are not sure how such approaches work outside deterministic MuJoCo environments.

---

### Official Review · Reviewer_XKs3 · 2021-11-03

**Correctness:** 4
**Technical Novelty And Significance:** 2
**Empirical Novelty And Significance:** 2
**Recommendation:** 5
**Confidence:** 4

**Details Of Ethics Concerns:**

I don't have ethics concerns of this paper.

**Main Review:**

Strength:
1.	the method is well motivated and derived
2.	Empirical study in various domains show diversity of the learned options and its usefulness in transfer settings.

Weakness:
1.	The method is incremental and seems trivial. The method is directly derived from the termination gradient theorem from Harutyunyan et al. (2019) with reward signal constructed based on mutual information between terminal state and option.

2.	Failed to mention and compare some relevant references, E.g., reference [1] also considered learned diversified options using pseudo-reward constructed based on divergence between action distribution of different options.

[1] Kamat & Precup, Diversity-Enriched Option-Critic, Neurips2020

[2] Ramesh et al., Successor Options: an option discovery framework for reinforcement learning, IJCAI2019

I would like to see some discussions of these references and experimental comparisons.

3. This work is mostly based on empirical study, the paper could be strengthened if some theoretical analysis, such as sample/computation complexity, convergence etc. can be provided.

Other comments:
In the abstract, “our experiments demonstrate … without rewards”,  you really mean “without environmental/extrinsic rewards”.
Proposition 2 seems simply a trivial extension of proposition 1.
In the experiment (5.1), why the policy over option $\mu(o|x)$ is fixed to be $1/|O|$, instead of learned?


**Summary Of The Paper:**

This paper proposed an algorithm, termed infomax Termination Critic (IMTC) to learn diversified options in RL. This algorithm learns termination conditions of options by maximizing mutual information between options and corresponding state transitions. The experiments demonstrate the IMTC algorithm learns diversified options and can be reused in various tasks.

**Summary Of The Review:**

In all, this paper studied an important problem in reinforcement, learning diversified options that can be reused in various tasks. Their empirical study is able to justify the diversify of the learned options and shows superior performance in transfer learning settings comparing to some baseline.  However, there are still some key references missing and should be compared against. Also, the method seems trivial extension of many existing methods.

---

> ### Author Response · Authors · 2021-11-23
> **Response to Reviewer XKs3**
>
> Thank you for your comments and suggestions.
> We will clarify our contribution and add some citations in the revised manuscript.
>
> > The method is incremental and seems trivial.
>
> We agree that the derivation of our method heavily depends on Harutyunyan et al. (2019) and looks incremental.
> However, still, we believe that the contributions made in this paper are beneficial for the community in that we provided simple, yet practical techniques for making termination gradient work in practice. In addition, we believe that our work effectively complements some weak points of VIC and the option critic. Here we note three points.
>
>   1. In section 3, we proposed to approximate $p_\mathcal{O}(o|x_s, x_f)$ instead of the option transition model $P^o(x_f|x_s)$, making the model easier to learn. We believe that this design choice is important to make termination gradients work in practice.
>
>   2. We confirmed that our method significantly clarifies the option policies learned with VIC. Since VIC depends on the learned option classifier $\hat{p}_\mathcal{O}(o|x_s, x_f)$, it cannot learn diverse options if $\hat{p}_\mathcal{O}$ is close between options. Therefore, each option has to get lucky of ending at different states so that $\hat{p}_\mathcal{O}$ becomes diverse. In this sense, we believe that IMTC can help reasonably many option learning methods based on state or trajectory classification.
>
>   3. As a small contribution, in section 4, we proposed upgoing GAE, which we confirmed effective in task adaptation experiments in the ablation study in Appendix C. The original advantage estimation in the option critic paper cannot fully benefit from N-step estimation, which we think is problematic especially in problems with sparse rewards. Our upgoing estimation resolves this problem.
>
> We will clarify these points in the revised manuscript.
>
> > Failed to mention and compare some relevant references
>
> Thank you for your recommendations.
> Diversity-Enriched Option-Critic paper seems to share the same goal as us, and we will cite it in the next revision.
> However, we only found the arxiv version of the paper and didn't find a published one in conference proceedings or journal, so we can be excused according to the [reviewer guide](https://iclr.cc/Conferences/2022/ReviewerGuide).
> Moreover, unfortunately, we decided not to compare our method with Diversity-Enriched Option-Critic, since we found two issues.
> 1. The paper says that their method maximizes divergence between two option policies $H(A^{o_1}; A^{o_2}|X)$, but the exact objective to optimize remains unclear.
> 2. [The official open-source implementation](https://github.com/anandkamat05/TDEOC/blob/master/baselines/Termination_DEOC/pposgd_simple.py) learns the termination functions by maximizing $\sum_{o_a, o_b \in \mathcal{O}} -\pi^{o_a}(A = \mu_{o_a} |x) \pi^{o_b}(A = \mu_{o_a} |x) \log(\pi^{o_a}(A = \mu_{o_a} |x) \pi^{o_b}(A = \mu_{o_a} |x))$, where $\mu_o$ is the mean of $\pi^o$. This is confusing to us.
>
> We appreciate your understanding of the difficulty we faced.
> Successor Options paper also looks interesting.
> Since it targets point-options, we will cite the paper in the discussion in section 6.1, where we discussed some point-options learning methods.
>
> > This work is mostly based on empirical study, the paper could be strengthened if some theoretical analysis, such as sample/computation complexity, convergence, etc. can be provided.
>
> Thank you for your suggestion. Although we do not have enough time for adding more theories to the paper, we will add more experimental analysis in the revised manuscript.
>
>
> > In the abstract, “our experiments demonstrate … without rewards”, you really mean “without environmental/extrinsic rewards”.
>
> Thank you for your correction. We will fix this in the revised manuscript.
>
> > In the experiment (5.1), why the policy over option $\mu(o|x)$ is fixed to be $\frac{1}{|\mathcal{O}|}$, instead of learned?
>
> This is because we assume the setting where the agent does not have prior knowledge of the future task distribution.
> In this case, since we are interested in pre-train options in a diverse way, using uniform distribution is the simplest way to train each option equally. We will clarify this point in the revised manuscript.

---

### Official Review · Reviewer_SWkr · 2021-11-07

**Correctness:** 3
**Technical Novelty And Significance:** 2
**Empirical Novelty And Significance:** 2
**Recommendation:** 5
**Confidence:** 3

**Main Review:**

The paper presents an extension of the termination critic idea, which is called the InfoMax Termination Critic. Specifically, the approach is different from Termination Critic as it involves computing the entropy over the final states as a function of the option and the start state (termination critic only computes the entropy of the final states as a function of the option).

The overall approach is clear and well-presented.

I do have a number of questions related to this approach and would like some clarifications from the authors:

1. The Fig 2 presented shows qualitatively how the terminations for options look for IMTC and Termination Critic. Would it be possible to demonstrate a similar visualization on the standard 4 room gridworld, identical to the one presented in Termination Critic’s paper? It would be much easier to make a comparison with the published result as opposed to reimplementing termination critic and demonstrating the differences on a new but smaller gridworld.

2. If I understand the approach correctly, the terminations for the options are discovered through IMTC but the policy for those options are discovered through Variational Intrinsic Control (VIC). Is my understanding correct?
This seems like a roundabout way of discovering options that are diverse. Why doesn’t the approach learn option-policies where the rewards for those options are obtained through the discovered termination functions?
Also, this makes it unclear if the performance gains are due to the discovered terminations or due to VIC. Perhaps adding an ablation to address this is necessary to make it clear.

3. Would it be possible to make Fig 3 more prominent? It is hard to understand what each of the option-policies are looking like. Also, how are the options being discovered for OC + VIC? I thought that OC requires extrinsic rewards (from tasks) to discover options and in the experiments considered in this paper, there are no extrinsic rewards (i.e., reward-free RL).

4. Because the approach is very much related to Termination Critic, I think it is important to include Termination Critic as a baseline in the large scale experiments (the plots in Fig 3 and Fig 5). Without this result, it is difficult to understand how the proposed work is better than its prior work.

5. What are the error bars in Fig 5?


**Summary Of The Paper:**

The paper presents an approach for learning diverse temporally extended and reusable options. It is based on the assumption that learning diverse options are generally useful for downstream tasks. Thus, the approach aims to discover options without making any assumptions about the downstream tasks. The main idea behind the approach is to discover options by maximizing the mutual information between the options and the corresponding state transitions and demonstrates the options discovered by this approach. Also, the paper demonstrates that these options are useful for faster learning in a downstream task.


**Summary Of The Review:**

Overall, I think the paper addresses an important problem in RL: discovering diverse options when there are no external tasks. The paper presents an adaptation of Termination Critic.
However, some of the design choices made by the approach does not seem to be reasonable. Specifically, why use Variational Intrinsic Control in combination with the proposed approach when there is a direct way of learning option-policies from the discovered termination functions. Also, using VIC in this way makes it unclear as to whether the performance gains are due to the proposed approach or due to VIC.

---

> ### Author Response · Authors · 2021-11-23
> **Response to Reviewer SWkr**
>
> Thank you for your thoughtful comments.
> We are now working on clarifying the motivation behind some decision choices in the revised manuscript.
>
> > Would it be possible to demonstrate a similar visualization on the standard 4 room gridworld, identical to the one presented in Termination Critic's paper?
>
> Thank you for your suggestion. We will add the visualization in Appendix C of the revised manuscript.
>
> > If I understand the approach correctly, the terminations for the options are discovered through IMTC but the policy for those options are discovered through Variational Intrinsic Control (VIC). Is my understanding correct?
>
> Yes, your understanding is correct. The policy is trained to maximize the cumulative VIC rewards. We will clarify this point in the revised manuscript.
>
> > Why doesn't the approach learn option-policies where the rewards for those options are obtained through the discovered termination functions?
>
> This is a great point. Although there can be many possible ways to compute rewards from terminating regions, we will add the results of additional experiments where we replace VIC rewards with a constant $0.01$. I.e., we give an agent the constant reward if an option ends. So far, our conclusion is that the constant rewards also produce qualitatively good options, but VIC is better in terms of task adaptation performance.
>
> > Would it be possible to make Fig 3 more prominent?
>
> Thank you for your suggestion. We will try to make the figure clearer in the revised manuscript.
>
> > Also, how are the options being discovered for OC + VIC?
>
> We also used the VIC reward for OC + VIC. So, in this case, OC's termination function $\beta$ is learned so that $\sum_i \gamma^i R_\textrm{VIC}$ is maximized. We will clarify this in the revised manuscript.
>
> > I think it is important to include Termination Critic as a baseline in the large scale experiments
>
> Unfortunately, as we shortly mentioned at the beginning of section 5, we failed to reproduce options with small entropy by our implementation of termination critic. A major difficulty we had is in training the option transition model $P^o(x_f|x_s)$ and the marginalized $P_o^\mu(x_f|x_s)$. More precisely, in our implementation, $P^o$ produced close values for each option, leading to similar termination regions per option. This failure motivated our design decision of approximating $p_\mathcal{O}(o|x_s, x_f)$ instead of $P^o$. We will clarify this point in the revised manuscript.
>
> > What are the error bars in Fig 5?
>
> We did not show error bars in Fig 5, but just showed scatter plots and learning curves regressed based on the resulting episodic returns.
> We will add 95% confidence intervals for both in the revised manuscript.

---

### Author Response · Authors · 2021-11-23
**Summary of the revision**

Dear all reviewers,

Thank you for your thoughtful comments.
We uploaded the revised manuscript. You can compare the manuscript at https://openreview.net/revisions/compare?id=UTTrevGchy&left=xKEC2gqb3n&right=z0VbgAP_Im&pdf=true.

Here, we briefly summarize the revision:

- We clarified our contributions in Section 1.
  - Some reviewers raised a concern about the novelty relative to VIC. Thus, we made the difference in our method with VIC and TC clearer.
- We added experiments with IMTC + RVIC in Section 5.
  - Related to the novelty concern relative to VIC, we pre-trained IMTC using RVIC rewards to show the wide applicability of IMTC.
  - Also, we added experiments with IMTC + constant rewards.
- We added experiments in Gridworld and discussed the limitation of our method in Section 5 and Appendix C.

We hope these changes would make sense to you and welcome any additional discussion or question.

---

### Decision · Program_Chairs · 2022-01-20

**Decision:**

Reject

**Comment:**

This paper proposes InfoMax Termination Critic (IMTC), a new approach for learning option termination conditions with the aim of discovering more diverse options. IMTC relies on a scalable approximation of the gradient of a mutual information objective with respect to the termination function parameters.

Reviewers liked the motivation and the simplicity of the approach. While there were some initial concerns regarding the similarity of IMTC and VIC, the authors did a good job of clarifying the differences and providing additional results in the rebuttal. While two reviewers raised their scores based on the rebuttal, this left reviewers split on whether to accept or reject the paper.

Given that the paper’s main contributions are evaluated empirically I based my decision on the strength of the evaluation. The main claim in the paper is that IMTC significantly improves the diversity of the learned options when combined with intrinsic control methods like VIC and RVIC. The main supporting evidence of this claim is a visualization of the option policies and termination probabilities reached by VIC and RVIC. There are several issues with this comparison:
* This is a poor visualization of the kind of option diversity the paper aims to obtain. Given that mutual information based objectives used by VIC, RVIC and IMTC aim to optimize diversity in the final states reached by the options, visualizing the distribution of final states or the trajectories produced by the options is more meaningful.
* The VIC and RVIC baselines are evaluated with a fixed option termination probability of 0.1 which biases the comparison in favor of IMTC because IMTC is able to choose when and where to terminate while VIC and RVIC with random termination get to control neither. Using fixed option duration with MI-based option discovery methods like VIC, DIAYN and RVIC is more standard and is known to produce options with very clear terminal state clusters which are well-separated for different options. Fixed option duration allows VIC and RVIC precise control of where they will terminate since option duration is fixed, hence it should have been included in the comparison.
* As mentioned in point 2 above, it is well established that VIC tends to learn options with well-clustered end states, especially in simple gridworld domains like in Figure 3 (see VIC, DIAYN and RVIC papers). The authors seem to obtain different qualitative results raising questions.

Overall, I don’t think the qualitative experiments show that IMTC is able to improve the diversity of options discovered by VIC or RVIC due to issues with how the experiments are done (random option duration for VIC and RVIC) and how the results are presented (visualizing action probabilities instead of final states). Given these concerns and the split among the reviewers I recommend rejecting the paper in its current form.